# Induced Neurodifferentiation of hBM-MSCs through Activation of the ERK/CREB Pathway via Pulsed Electromagnetic Fields and Physical Stimulation Promotes Neurogenesis in Cerebral Ischemic Models

**DOI:** 10.3390/ijms23031177

**Published:** 2022-01-21

**Authors:** Hee-Jung Park, Ju-Hye Choi, Myeong-Hyun Nam, Young-Kwon Seo

**Affiliations:** Department of Medical Biotechnology, Dongguk University, Goyang-si 10326, Korea; ireneparkhj@gmail.com (H.-J.P.); jooh031919@gmail.com (J.-H.C.); iis05047@naver.com (M.-H.N.)

**Keywords:** cerebral ischemia stroke, pulse electromagnetic fields, inflammatory cytokines, neural differentiation

## Abstract

Stroke is among the leading causes of death worldwide, and stroke patients are more likely to live with permanent disabilities even after treatment. Several treatments are being developed to improve the quality of life of patients; however, these treatments still have important limitations. Our study thus sought to evaluate the neural differentiation of human bone marrow mesenchymal stem cells (hBM-MSCs) at various pulsed electromagnetic field (PEMF) frequencies. Furthermore, the effects of selected frequencies in vivo were also evaluated using a mouse ischemia stroke model. Cell proliferation decreased by 20% in the PEMF group, as demonstrated by the 3-[4,5-dimethylthiazol-2-yl]-2,5-diphenyl tetrazolium bromide (MTT) assay, and lactate dehydrogenase (LDH) secretion increased by approximately 10% in an LDH release assay. Fluorescence-activated cell sorting (FACS) analysis demonstrated that CD73 and CD105 were downregulated in the PEMF group at 60 Hz. Moreover, microtubule-associated protein 2 (MAP-2) and neurofilament light chain (NF-L) were upregulated in cell cultures at 60 and 75 Hz. To assess the effects of PEMF in vivo, cerebral ischemia mice were exposed to a PEMF at 60 Hz. Neural-related proteins were significantly upregulated in the PEMF groups compared with the control and cell group. Upon conducting rotarod tests, the cell/PEMF group exhibited significant differences in motor coordination at 13 days post-treatment when compared with the control and stem-cell-treated group. Furthermore, the cell and cell/PEMF group exhibited a significant reduction in the expression of matrix metalloproteinase-9 (MMP-9), tumor necrosis factor-alpha (TNF-α) and interferon-gamma (IFN-γ) in the induced ischemic area compared with the control. Collectively, our findings demonstrated that PEMFs at 60 and 75 Hz could stimulate hBM-MSCs neural differentiation in vitro, in addition to promoting neurogenesis to enhance the functional recovery process by reducing the post-stroke inflammatory reaction.

## 1. Introduction

Despite current declines in the mortality rates associated with neurological diseases, stroke continues to be the second-leading cause of death and long-term disability worldwide [1,2]. Stroke had been traditionally viewed as a disease of the elderly. However, there has been a significant increase in the incidence of stroke in young and middle-aged adults [3]. To ameliorate the effects of strokes, researchers recently reported a new and effective treatment based on stem cell transplantation [4]. The use of stem cells in clinical medicine has garnered tremendous momentum over the last decade, and adult bone marrow is currently the most well-characterized source of adult stem cells [5]. Bone marrow mesenchymal stem cells (BMSCs) originating from the bone marrow (BM) can differentiate into other cell types in different organs [6,7]. Many studies have demonstrated that BMSCs can differentiate into neural lineage cells when induced by key factors, thus aiding in the treatment of a variety of neurological diseases [8,9,10]. Although various approaches have been proven to restore the functions of damaged brain tissues poststroke, the ability of current therapies to rehabilitate stroke patients remains limited. Therefore, novel treatments are needed to minimize side effects and improve the prognosis of stroke patients [11,12,13].

Several studies have successfully employed pulsed electromagnetic field (PEMF) technology as an alternative non-invasive treatment method in a variety of fields. Particularly, electromagnetic fields (EMFs) have been found to enhance peripheral nerve regeneration (1 mT, 50 Hz, 1 h/day for 3 weeks) in rat models; increase myofibroblast populations; contribute to wound closure during diabetic wound healing (5 mT, 50 Hz, 1 h/day for 21 days); and promote chondrocyte proliferation, matrix synthesis, and chondrogenic differentiation in rat models [14,15]. Another study also reported that EMFs (1 mT, 50 Hz, 3.5 h/day for 12 days) enhances the proliferation and neuronal differentiation of hippocampal neural stem cells (NSCs) in adult male mice [16]. Furthermore, other studies have found that EMF therapy (repeating at 2 Hz, 15 min/two times daily for 21 days) had a positive impact on poststroke recovery, as demonstrated by a reduction in ischemic infarction size in an ischemia mouse model. Furthermore, EMF (0.4 MT, 50 Hz, 24 h/day for 60 days) partially improved the cognitive and clinicopathologic symptoms of an Alzheimer’s disease rat model [17,18].

The purpose of this study was to identify the optimal PEMF frequency for neurodifferentiation at various intensities in vitro, and to evaluate the wound-healing efficacy in a mouse stroke model. Additionally, to gain insights into the poststroke wound healing process, our study assessed whether stem cells and electromagnetic fields could synergistically aid in the treatment of ischemic stroke.

## 2. Results

### 2.1. Morphological and Toxic Effects of PEMF Exposure on hBM-MSCs

The hBM-MSCs were cultured in a neural-induction medium and exposed to PEMF (Figure 1). After three days of culture, cells with long bipolar and dendrite-like cytoplasmic projections were found in the 60 and 75 Hz PEMF groups, but not in the control group. To test the effect of the PEMF on the growth of hBM-MSCs, cell counting was performed using a Scepter automated cell counter after three days of cell culture. The total cell numbers in the treated PEMF groups tended to be lower than that of the control group. The cell activity of induced hBM-MSCs was analyzed using the MTT assay after three days. The cell activities of the PEMF groups increased significantly at 60 and 75 Hz compared to the control group. Additionally, an LDH assay was performed using 3 days post culture cell medium supernatants to evaluate cellular damage. As shown in Figure 1, there was a significant difference in LDH between the control and the 60 and 75 Hz PEMF groups. Therefore, our findings suggested that PEMF exposure affected hBM-MSCs morphology and proliferation.

### 2.2. FACS Analysis

The expression levels of CD73 and CD105 were evaluated three days after culture using FACS analysis. In the control group, the expression levels of CD73 and CD105 were 92% and 83%, respectively. As shown in Figure 2, PEMF treatment at 30, 45, 60, and 75 Hz resulted in CD73 expression levels of 91%, 82%, 83%, and 83%, respectively. Moreover, the same frequencies resulted in CD105 expression levels of 82%, 83%, 68%, and 74%. Interestingly, CD73 expression tended to decrease at 45, 60, and 75 Hz, whereas CD105 decreased at 60 and 75 Hz compared to the control group. These findings indicated that PEMF exposure increased differentiation. Furthermore, the transformed surface marker expression of hBM-MSCs was reduced, resulting in the conversion of these stem cells into specific cell types.

### 2.3. Immunohistochemical Staining

As shown in Figure 2, immunohistochemical staining was performed using anti-MAP-2 as a neuronal marker to confirm the neural differentiation of the hBM-MSCs. Our findings indicated that most of the cells in the 60 and 75 Hz PEMF groups expressed MAP-2 after three days. However, MAP-2 was not expressed in the control and 30 Hz PEMF groups. We also verified the expression of NF-L, a neuronal-specific protein, through immunofluorescence staining. Interestingly, the expression of NF-L increased in the 45, 60, and 75 Hz PEMF groups. Taken together, our findings indicated that PEMF at 60 and 75 Hz frequencies greatly induced the differentiation of hBM-MSCs to neural-like cells.

### 2.4. RT-PCR and Western Blotting

After three days of culture, the expression of neural-related markers at the mRNA and protein levels were quantified via RT-PCR and Western blotting analyses. The mRNA expression levels of the neural development genes Neuro D1, MAP-2, Tau, MBP, DCX, and NF-L were determined using total mRNA obtained after three days. As shown in Figure 3A, neural markers were upregulated in the PEMF groups compared to the control group. Furthermore, the 60 Hz PEMF groups showed a significant increase in neural-related gene expression compared to the other PEMF groups. Our experiments also confirmed the mRNA expression of Wnt3α and β-catenin, which have essential roles in neural differentiation and proliferation. Both Wnt3α and β-catenin increased in the PEMF groups compared to the control group. Furthermore, we analyzed the expression of ERK, p-ERK, CREB, p-CREB, and β-catenin signaling by Western blotting after exposure to the PEMF for three days. As shown in Figure 3B, the phosphorylation of ERK and CREB and expression of β-catenin after exposure to the PEMF increased compared to the control group. Particularly, p-ERK, p-CREB, and β-catenin signaling were highly stimulated in the 60 and 75 Hz PEMF groups. Therefore, our results indicated that the expression of genes and proteins associated with neuronal differentiation were upregulated in the 60 and 75 Hz PEMF groups.

### 2.5. Effect of PEMF Exposure on Motor Function in a Mouse Ischemia Model

To assess whether PEMF treatment affected the behavior and motor coordination of mice with cerebral ischemia, a rotarod test was conducted daily after hBM-MSC transplantation. All data are presented as a percentage relative to the rotarod results of normal animals before treatment. The graph in Figure 4 shows the results of the motor function recovery assays. A total of 10 mice were used for each experimental group, and the animals that died during the experimental period after surgically inducing brain ischemia were excluded from this experiment. Therefore, a total of 5, 5, and 9 mice were ultimately evaluated for the control, cell, and cell/PEMF groups, respectively. The results reported herein correspond to those observed at 9 days, which is when significant differences were observed. No significant differences were observed in the control group, with an average value of 24.60% ± 11.20 15 days after surgery (Table 1). In contrast, the cell and cell/PEMF groups exhibited significant improvements at 9 days post-transplantation. The cell group exhibited a large increase of 43.76% ± 14.25 at 13 days, 55.49% ± 18.09 at 14 days, and 63.76% ± 13.70 at 15 days after hBM-MSC transplantation. Furthermore, the cell/PEMF group showed a significant difference of 68.07% ± 24.29 at 13 days, 74.45% ± 12.63 at 14 days, and 74.97% ± 13.70 at 15 days after hBM-MSC transplantation and PEMF exposure. Therefore, rapid recovery of motor function was observed in the cell/PEMF groups with hBM-MSC transplantation. We thus concluded that PEMF exposure could enhance functional recovery in the ischemia stroke model compared to the other two groups.

### 2.6. Evaluation of Neural Protein Expression and Histological Analysis after PEMF Exposure in the Mouse Stroke Model

All mice were sacrificed after 15 days of treatment and endured the treatments without significant weight loss. The expression levels of the neural development proteins Tau, Neuro D1, and Nestin in brain tissue were evaluated by Western blotting and the ImageJ software. There was a significant increase in the expression level of neural proteins in the cell group and the cell/PEMF group compared to the control group (Figure 5). In the cell group, Tau increased by 7.6-fold, Neuro D1 by 1.7-fold, and Nestin by 4.7-fold compared to the control group. In the cell/PEMF group, Tau increased by 13.4-fold, Neuro D1 by 2.1-fold, and Nestin by 3.2-fold compared to the control group. Furthermore, Tau and Nestin expression was significantly different in the cell and cell/PEMF groups, and significant differences in Neuro D1 expression were observed in the cell/PEMF group. Therefore, our findings indicated that PEMF treatment promoted neurogenesis of hBM-MSCs in the mouse ischemia model.

Cerebral infarction lesions in brain tissues were confirmed via H&E staining. Prussian Blue staining was conducted to confirm that the injected cells had successfully migrated to the damaged brain area. As illustrated in Figure 6, blue-colored cells were identified in the cell and cell/PEMF groups, but not in the control group. Additionally, to investigate the healing effect of hBM-MSCs, the cerebral infarction lesion was stimulated by PEMF in the mouse ischemia model, after which the expression levels of Neuro D1, NF, and BDNF were evaluated via immunohistochemical staining. The expressions of Neuro D1, NF, and BDNF in the cell group and the cell/PEMF group increased compared to the control group. Particularly, these proteins were more robustly expressed in the cell/PEMF group than in the control group. As illustrated in Figure 7, we also investigated whether cell/PEMF treatment affected the expression of the inflammatory intermediaries MMP-9, TNF-α, and IFN-γ in the mouse ischemia model. We observed significant reductions in the expression of MMP-9, TNF-α, and IFN-γ compared to the control in areas surrounding the wound sites in the cell and cell/PEM groups.

## 3. Discussion

In this study, neural differentiation of hBM-MSCs was induced via PEMF treatment to assess whether this could serve as a therapy to aid in poststroke recovery, as well as to demonstrate the effect of electromagnetic fields on the post-transplantation healing effects of hBM-MSCs in mouse ischemic models. First, we examined the cell morphological changes and toxicity of neural-induced hBM-MSCs after exposure to various PEMF intensities. Previous studies have comprehensively characterized the effects of neural differentiation media on hBM-MSCs [19,20,21,22]. To determine the effect of PEMF on hBM-MSCs and neural induction, we used a basic neural differentiation medium with forskolin and insulin. As shown in Figure 1A, morphological changes were identified in both the control and PEMF groups, which were unlike the original spindle shape of bone marrow mesenchymal stromal cells. Neural-like cells with elongated cytoplasm were observed in the 60 and 75 Hz PEMF groups. The viability and cytotoxicity of the changed neural-like cells were also evaluated using the MTT and LDH assays. Interestingly, the results of our MTT analysis indicated that PEMF exposure increased cell activity, whereas the results of our LDH analysis were indicative of high levels of cell stress due to defects in cell membranes. This was likely because PEMF stimulation promotes cell differentiation but reduces the cell membrane thickness.

FACS was then performed to assess the differentiation of mesenchymal stem cells (MSCs) (Figure 2). Low CD73 and CD105 expression levels were observed in the PEMF groups compared with the control. Particularly, there was a significantly lower expression in the 60 Hz PEMF group compared to the other PEMF groups. Many studies have reported that multipotent MSCs have a positive expression of 99% for CD73, CD90, and CD105, which are typical cell surface markers [5,23,24]. In our study, expression of CD75 and CD105 decreased in the 60 Hz PEMF group compared to the control group, suggesting that PEMF exposure accelerated the differentiation of hBM-MSCs. Additionally, the cells were subjected to immunohistochemical and immunofluorescence analysis to evaluate the expression of typical neural markers such as MAP-2 and NF after PEMF exposure (Figure 2B). MAP-2 was expressed during the neuronal differentiation of neural precursor cells. Interestingly, our data showed a considerable increase in the expression of both MAP-2 and NF-L in the 60 and 75 Hz PEMF groups. This suggested that PEMF increased the differentiation of hBM-MSCs to neural-like cells.

Our results also confirmed the neural differentiation of hBM-MSCs by characterizing the expression of mRNA and proteins as shown in Figure 3. Neural-related genes such as Neuro D1, MAP2, Tau, MBP, DCX, and NF-L were highly expressed in the PEMF groups. Therefore, our data clearly demonstrated that PEMF promoted neural differentiation of hBM-MSCs via the expression of neural-related markers. Furthermore, Wnt3α and β-catenin were significantly upregulated in the PEMF groups compared to the control group (Figure 3A). Previous studies have reported that Wnt3α is a cysteine-rich, lipid-modified protein that plays a major role in various processes during cell development, including proliferation and differentiation. In turn, β-catenin is an essential component of the Wnt3α signaling system [25,26]. Related studies have shown that activation of the Wnt3α and β-catenin pathway promotes neuronal differentiation and regulates Wnt3α and β-catenin during mesenchymal stem cell therapy [27,28]. Other studies have demonstrated that the activation of Wnt3α signaling is a key pathway involved in the improvement of neural regeneration after injury, as the Wnt3α pathway promotes the differentiation of neural stem cells and facilities tissue repair in vivo [29]. As shown in Figure 3A, the PEMF groups exhibited a significant increase in Wnt3α and β-catenin activation, which may have accelerated hBM-MSC neurogenesis.

This study evaluated the effect of PEMF on the phosphorylation of CREB and ERK through Western blotting (Figure 3B). Both the 60 and 75 Hz PEMF groups exhibited increases in ERK phosphorylation. ERK signaling in neural precursors is crucial for the stimulation of adult neurogenesis [30]. ERK, a member of the mitogen-activated protein kinase (MAPK) family, is the upstream kinase of CREB, and phosphorylated ERK activates the phosphorylation of CREB [31,32]. Previous studies have reported that CREB regulates neurotrophic factor-induced cell survival, and activation of this pathway leads to neuronal differentiation and neurite outgrowth [33,34,35]. Related studies have demonstrated that brain-ischemia-induced neurogenesis involves stimulation of receptor tyrosine kinases by induction of growth factors that stimulate phosphatidylinositol 3-kinase (PI3K)/Akt and the ERK pathway [30]. Park et al. reported that EMF-induced CREB phosphorylation could promote neural differentiation in BM-MSCs in vitro [36].

As shown in Figure 4, this study also assessed behavioral outcomes after transplantation using the rotarod test. The rotarod test is a widely used method to evaluate motor coordination in rodents. Our results demonstrated that the cell/PEMF group had a markedly reduced falling latency. Additionally, the animals in the cell/PEMF group showed significant improvements nine days after transplantation compared to the control and cell groups, as shown in Table 1. The animal test was originally conducted using 10 mice per group; however, the cell/PEMF group had a higher survival rate than the other experimental groups. Although additional experiments are needed, our findings suggested that recovery from cerebral infarction impacted the survival rate of animals.

As shown in Figure 5, the in vivo expressions of neural-related protein levels of Tau, Neuro D1, and Nestin were increased. Our study confirmed that the expression of Tau was significantly increased in the cell/PEMF group. Tau protein is primarily expressed in the brain. This protein has six isoforms produced by alternative mRNA splicing of the microtubule-associated Tau gene, which comprises 16 exons on chromosome 17q21.31. The primary physiological function of the Tau protein is to stabilize microtubule networks within neurons [37]. Furthermore, the dynamic microtubule network provided by the Tau protein is crucial for the proper migration of new neurons. For example, a previous study demonstrated that Tau knockout resulted in a severe reduction in adult neurogenesis in mice [38]. Additionally, Nestin was originally described as a neural stem/progenitor cell marker and was observed in undifferentiated central nerve system (CNS) cells during development, as well as in the normal adult CNS. Previous studies have reported that Nestin is upregulated in injured adult tissue, where its expression is thought to contribute to tissue regeneration [39]. To confirm this result, we performed immunohistochemical analysis to confirm the positive expression of neural proteins Neuro D1, NF, and BDNF (Figure 6). Neuro D1 was reported to be essential for adult neurogenesis and was shown to induce terminal neuronal differentiation [40,41,42]. BDNF is known to play an important role in regulating neuronal migration, differentiation, synaptic remodeling, and survival in the mature nervous system [14]. Additionally, BDNF is also a key factor involved in hippocampal cell proliferation and neuronal differentiation [43]. Prussian Blue staining was conducted to verify the location of the injected cells in the mouse tissues. The injected cells were located throughout the ischemic region of the brain. In previous studies, MSC therapy contributed to axonal remodeling around the ischemic lesion, which coincided with improved functional recovery. The therapeutic effects of MSC transplant were attributed to the secretion of factors that reduced levels of axonal growth inhibitors and promoted growth and neurogenesis [44,45,46]. Therefore, our data indicated that PEMF stimulated the implanted hBM-MSCs, thus enhancing neural tissue generation.

Cytokines play an important role in the communication between cells, and they regulate survival, growth, differentiation, and effector functions [47]. MMPs are a family of zinc-dependent endoproteases with multiple roles in tissue remodeling and degradation of various proteins in the extracellular matrix (ECM). MMPs promote cell proliferation, migration, and differentiation, and could play a role in cell apoptosis, angiogenesis, tissue repair, and immune response [48]. MMP-9 is secreted by macrophages, and its constant expression throughout a wound could contribute to the loss of neuronal cells and damage to the surrounding tissue. MMP-9 increases the permeability of the blood–brain barrier (BBB), thus facilitating the infiltration of leukocytes into the CNS, degrading the myelin layer, and consequently resulting in neuronal damage [49]. Some researchers have shown that PEMF significantly decreased inflammatory IL-1β and MMP-9 in the ischemic stroke brain [50]. MMP expression was increased in the early stages of wound healing, but its expression was reduced at the later stage of the wound-healing process to prevent secondary damage. The proinflammatory cytokine TNF-α is essential to control brain immune response and protect neuronal cells against pathogens. However, its overexpression has been linked to neuron damage and adverse effects on other cells in the CNS [51]. Other studies have suggested that TNF-α can induce and enhance the inflammatory reaction via the activation of glia and blood cells such as macrophages and neutrophils. Indeed, the levels of proinflammatory cytokines such as TNF-α, IL-1β, and IL-6 were significantly elevated in the early stage after brain injuries [52]. Here, we reported the effect of high-intensity PEMF on an in vivo stroke model. Our data indicated that TNF-α significantly decreased in the 60 Hz cell/PEMF group (10 MT for 30 min) after cell transplantation in the mouse stroke model. IFN-γ signaling appeared to play a key role in stroke-induced neurodegeneration. Inhibiting IFN-γ signaling resulted in a reduction in infarction volume and size. Therefore, selectively targeting the IFN-γ signaling pathway is a potential treatment for stroke [53]. Our model inhibited the release of the proinflammatory cytokine IFN-γ in the injured brain wound area in mice. Furthermore, our findings suggested that stem cell treatment coupled with 60 Hz PEMF treatment at high intensity could decrease the production of inflammatory cytokines, which could aid in the post-stroke recovery process.

## 4. Materials and Methods

### 4.1. Culture and Neural Induction of hBM-MSCs

The hBM-MSCs were purchased from ATCC (ATCC PCS-500-012). The cells were cultured in a nonhematopoietic (NH) stem cell medium (Miltenyi Biotech, Bergisch-Gladbach, Germany) containing 1% penicillin/streptomycin (Welgene, Daejeon, Korea) at 37 °C in a 5% CO_2_ humidified atmosphere. The culture medium was changed twice a week. The cells were passaged before reaching confluence and were used after five passages. To induce neural differentiation, the cells were cultured in neural-induction media with a formulation based on previous publications with some modifications [21]. The neural-induction media contained the following: DMEM/F12 (Gibco-BRL, Grand Island, NY, USA) containing 5 mM potassium chloride (KCl), 2 μM valproic acid, 1 μM hydrocortisone, 10 μM forskolin (Sigma-Aldrich, St. Louis, MO, USA), and 5 μg/mL insulin (Welgene, Daegu, Korea). The medium was changed every two days.

### 4.2. Design of the Magnetic Generator

The PEMF was generated using a pair of Helmholtz coils. This device produced a region of nearly uniform magnetic fields. The stimulation unit was designed to handle a pair of identical coils 30 cm in diameter assembled in a Helmholtz configuration. A continuous PEMF (Bm = 10 mT, F = 30 Hz, 45 Hz, 60 Hz, 75 Hz) was used in this study. Alternating current was used to generate the electromagnetic fields with the coils. The current in the coil was generated using the COMSOL 3.4 simulation model (COMSOL, Inc., MA, USA), and the intensity was measured using a TM-701 Gaussmeter (Tesla meter TM-701, KANETEC, Tokyo, Japan). The PEMF system was placed in an incubator (MCO 175, SANYO, Japan) at 37 °C in 5% CO_2_ humidified air.

### 4.3. Exposure of hBM-MSCs to Pulsed Electromagnetic Fields

The Helmholtz coil was placed in the cell incubator, and the field was set to various continuous PEMF frequencies, including 30 Hz, 45 Hz, 60 Hz, and 75 Hz, each with a field intensity of 10 mT, for 30 min per day for three days. The hBM-MSCs (1.0 × 10^4^ cells/mL) were then placed in the base of the PEMF system. The control cells were incubated under the same experimental conditions as the group without the PEMF. Changes in the cell morphology under different culture conditions were observed using a microscope (Eclipse TS100, Nikon, Japan). Cell morphology photographs were taken three days after PEMF exposure.

### 4.4. hBM-MSC Growth Assay

Cell growth was determined via the MTT assay after three days of cell culture coupled with PEMF exposure. The cells (1.0 × 10^4^ cells/mL) were incubated in 0.5 mg/mL MTT-supplemented cell culture medium at 37 °C and 5% CO_2_ for 2 h. All experiments were performed in triplicate (*n* = 3). The purple-colored formazan derivative formed during the active cell metabolism was eluted and dissolved in 1 mL dimethyl sulfoxide (DMSO; Sigma-Aldrich, St. Louis, MO, USA). The absorbance was measured at a 570 nm wavelength. LDH activity was measured using an LDH-LQ kit (Asan Pharmaceutical Inc., Seoul, Korea). After three days of culture, 20 μL medium aliquots and 50 μL of working solution were thoroughly mixed and incubated in the dark at room temperature for 30 min. The reaction was terminated through the addition of a stop solution (1N HCl), and the absorbance was measured at a 570 nm wavelength.

### 4.5. Fluorescence-Activated Cell Sorting (FACS) Analysis

Cells were characterized by using a fluorescence-activated cell sorting (FACS) assay after three days of cell culture with PEMF exposure. CD 73 monoclonal antibodies (Invitrogen, Product # 12-0739-42, USA) and CD 105 (Endoglin) monoclonal antibodies (Invitrogen, Product # 12-1057-42, USA) were used to detect human antigens. Cells were incubated with fluorescein isothiocyanate (FITC) or with 200 μL of PBS and phycoerythrin (PE)-conjugated antibodies for 20 min at room temperature. The fluorescence intensity was examined using a flow cytometer (FACScan; BD Bioscience), and the data were analyzed using the CellQuest software (BD Biosciences).

### 4.6. Reverse Transcription-Polymerase Chain Reaction (RT-PCR)

Total cellular RNA was extracted using the TRIzol reagent (Invitrogen, Waltham, MA, USA) according to the manufacturer’s instructions. Total RNA concentrations and purity were measured using a NanoDrop spectrophotometer (Thermo Fisher Scientific, Waltham, MA, USA). A total of 2 μg of RNA was used for cDNA synthesis. Reverse transcription (RT) was conducted using the Advantage RT-PCR kit (Clontech, Palo Alto, CA, USA). Primers for RT-PCR were purchased from Bioneer (Deajeon, Rep, Korea) for the following genes: GAPDH, Neuro D1, MAP2, Tau, MBP, DCX, NF-L, Wnt, and β-catenin. All primer sequences are summarized in Table 2. The products of the PCR reaction were visualized via electrophoresis in 1.5% agarose gels stained with SYBR Safe DNA Gel Stain (Invitrogen, Waltham, MA, USA). Band images were obtained with a ChemiDoc XRS+ gel imaging system (Bio-Rad, Hercules, CA, USA). Quantitative analysis of the RT-PCR band images was conducted using the ImageJ software (National Institutes of Health, Bethesda, MD, USA).

### 4.7. Preparation of Experimental Animals

Experiments were performed using C57BL/6 male mice (Samtako Bio Korea, Osan, Korea) (8 weeks old, *n* = 30). The experimental protocol was approved by the Institutional Animal Care and Use Committee of Dongguk University (IACUC-2019-023-1). To create the stroke model, ischemic surgery was conducted as described by Choi et al. [54,55,56]. Mice were anesthetized via intraperitoneal (i.p.) injection of tiletamine/zolazepam (Zoletile; Virbac Lab., Carros, France) and xylazine (Rompun; Bayer, Seoul, South Korea). The photosensitive dye Rose Bengal (Sigma-Aldrich, St. Louise, MO, USA; 0.1 mL of a 10 mg/mL solution in sterile saline) was injected into the penile vein 5 min prior to illumination. The brain was illuminated for 20 min by placing a fiberoptic bundle of a cold light source (KL 1500 LCD, Schott, Germany) over the right frontal cortex with a focus at 0.5 mm anterior to the bregma and 2.5 mm lateral from the midline. All mice ingested a mixture of gentamycin sulfate and cephradine (23.33 mL kg^−1^ day^−1^). The lesioned animals were then randomly assigned to different experimental groups: control group (stroke without stem cell injection, *n* = 10), cell group (stroke with stem cell injection, *n* = 10), and cell/PEMF group (stroke with stem cell injection and PEMF exposure, *n* = 10). Fifth passage hBM-MSCs were used for transplantation, and these cells were implanted into the mice without inducing neuronal differentiation. Neural-induced hBM-MSCs were also identified in vitro (10 mT, 60 Hz and 75 Hz, 30 min). Animal testing was conducted under low-frequency conditions to ensure animal safety. The mice were exposed to PEMF for 30 min at a 10 mT intensity and a 60 Hz frequency. The hBM-MSCs were administered by inserting magnetic nanoparticles into the cell cultures to evaluate their movement and location in the animal prior to injection. Water-dispersible and biocompatible magnetic iron oxide (Fe_3_O_4_) nanoparticles (MNPs) were prepared as described in a previous study [56]. The hBM-MSCs (1 × 10^5^ cells) including MNPs were injected in saline via the penis, and the cell/PEMF group was exposed to PEMF (F = 60 Hz, 10 mT) for 30 min per day for 15 days.

### 4.8. Western Blot Analysis

The hBM-MSCs were solubilized in sample buffer containing 0.1 mg/mL bromophenol blue (Sigma-Aldrich, St. Louis, MO, USA), 2% SDS, and 10% glycerol in Tris-HCl, pH 6.8 (BIOSESANG, Gyeonggi-do, Korea), and boiled at 100 °C for 5 min. The protein concentrations of total lysates were determined via the BCA Protein Assay (Thermo Scientific, Rockford, IL, USA). 2-Mercaptoethanol (Sigma-Aldrich, St. Louis, MO, USA) was added to the sample buffer to a 10% concentration. The same amount of denatured protein (20–40 μg) was resuspended in a 10% SDS-PAGE gel and transferred to a nitrocellulose membrane. Membranes were blocked with 5% skim milk in TBS-T (pH 7.4) for 30 min at room temperature, followed by overnight incubation with antibodies at 4 °C. The following primary antibodies were used: β-actin, p-ERK, β-catenin (1:1000; Abcam, Cambridge, UK), and p-CREB (1:1000; Cell Signaling Technology, Beverly, MA, USA). Antibodies were appropriately diluted in TBS-T containing 0.05% BSA. After washing, membranes were incubated with antimouse or antirabbit secondary antibodies (1:1000; Abcam, Cambridge, UK) diluted in 5% skim milk in TBS-T for 2 h at room temperature. Band images were obtained using a ChemiDoc XRS+ gel imaging system (Bio-Rad, Hercules, CA, USA) using ECL solution (Thermo Fisher Scientific, Waltham, MA, USA). Quantitative analysis of Western blot band images was conducted using the ImageJ software (National Institutes of Health, Bethesda, MD, USA).

Brain tissues were obtained using a biopsy punch 15 days after the animal experiments. A 3 mm diameter and 3 mm depth orifice were punched into the right front core, which is where the cerebral infarction region was made (anterior 1.5 mm to the bregma and 2.5 mm lateral from the midline). β-actin, Tau (1:1000; Abcam, Cambridge, UK), Neuro D1, and Nestin (1:1000; Cell Signaling Technology, Beverly, MA, USA) were used as primary antibodies, and Western blotting was performed as described above.

### 4.9. Histological and Immunohistological Analysis

Cells on cover slides were fixed with 10% formalin for 30 min and washed with 10 mM Tris-HCl, pH 7.2. These cover slides were incubated with the respective monoclonal antibodies against anti-MAP-2 (1:100; Abcam, Cambridge, UK) for 24 h, followed by development using the EnVision Plus reagent and Mayer’s hematoxylin as a counterstain. The hBM-MSCs were then fixed with 4% paraformaldehyde in PBS for 15 min. PBS solution containing 1.5% bovine serum albumin (BSA) was used for cell blocking. Cells were permeabilized using 0.5% Triton X-100 for 16 h at 4 °C mixed with the primary antibodies, which included anti-Neurofilament (1:100; Cell Signaling Technology, Beverly, MA, USA). The cells were washed in PBS for 10 min and incubated with antirabbit IgG-Alexa Fluor 488 conjugate or antimouse IgG-Alexa Fluor 555 conjugate (1:100; Cell Signal Technology, Beverly, MA, USA) for 1 h at room temperature. The coverslips were counterstained with Vectashield mounting media containing 4‘,6-diamidino-2-phenylindole (DAPI; Vector Laboratories, Burlingame, CA, USA) for fluorescence and mounted on microscope slides. The cells were examined using a model LSM510-meta confocal laser scanning microscope (Carl Zeiss, Jena, Germany).

For hematoxylin and eosin (H&E) staining, the brain tissue was fixed in 4% paraformaldehyde for 72 h, dehydrated in ethanol, embedded in paraffin, and cut into 5 μm thick sections. For Prussian blue staining, brain tissues were washed with PBS and incubated in 5% potassium ferrocyanide in 5% hydrochloride acid. After 30 min, the slides were washed with PBS, counterstained, and examined under a light microscope. For immunohistochemical (IHC) staining, brain sections were incubated with anti-Neuro D1, anti-NF, anti-BDNF, anti-MMP-9, anti-IFN-γ, and anti-TNF-α antibodies (1:100; Abcam, Cambridge, UK). The sections were quenched with hydrogen peroxide, unmasked with hyaluronidase, and blocked in 10% normal donkey serum, then incubated with primary antibody for 1 h, washed, incubated with biotinylated secondary antibody, visualized with the Vectastain ABC staining kit (Vector Laboratories, Burlingame, CA, USA), and developed in 3,3-diaminobenzidine.

### 4.10. Rotarod Test

The rotarod test was used to measure motor coordination and fatigue prevention in mice after brain ischemia treatment. Only animals that were able to stay in the rotating rod for more than 180 s were selected for this study. The experimental mice were trained daily for three days before the surgery for the rotarod (3 cm in diameter, 15 rpm) test. To evaluate animal behavior, the rotarod test was performed every day for 15 days after transplantation. The mice were placed on the rotating rod at 15 rpm. The test consisted of three blind trials separated by 10 min intervals. The time spent on the rod was calculated after the mice fell from the test apparatus. 

### 4.11. Statistical Analysis

All data were analyzed using the SPSS 10.0 software (SPSS, Chicago, IL, USA) and are reported as mean ± SD. The differences between the groups were assessed using Student’s *t*-test. The results of the animal tests were analyzed using a one-way analysis of variance (ANOVA) followed by Tukey’s post hoc test. A *p*-value < 0.05 was considered statistically significant.

## 5. Conclusions

Stroke is among the leading causes of serious disability and death worldwide, and its occurrence has been rising for decades due to recent demographic shifts and aging populations. MSCs have been proven to be safe alternatives for stroke treatment. However, their efficacy as a poststroke therapy continues to be limited. Recently, PEMF (a non-invasive treatment that can be safely used in the long term) has been successfully used as a regulation therapy for a wide range of clinical conditions. In this study, physical PEMF treatment was used to minimize stroke side effects to improve the limitations of hBM-MSC therapy. Our findings confirmed that PEMF induced phosphorylation of ERK and CREB in vitro to promote hBM-MSC neurodifferentiation. Additionally, it was confirmed that the injected hBM-MSCs moved to the wound of the mouse stroke model. Using immunochemical staining, we confirmed that the injected cells could relieve inflammation of the ischemic area. In conclusion, we identified positive effects such as neurodifferentiation, motor function recovery, and nerve regeneration induced by inflammatory cytokine reduction in the PEMF-treated experimental groups with both in vitro and in vivo hBM-MSC treatment. Although additional studies are needed to clarify the exact therapeutic mechanisms of PEMF/hBM-MSC treatment, our findings provide a basis for the development of novel stroke rehabilitation strategies.

## Figures and Tables

**Figure 1 ijms-23-01177-f001:**
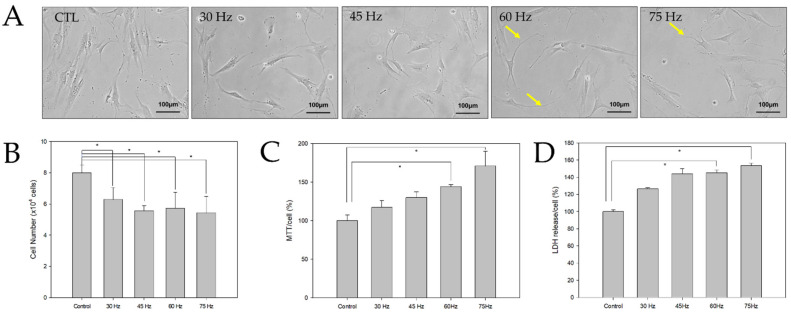
(**A**) Morphological changes of hBM-MSCs exposed to PEMF for 3 days (bar = 100 μm). (**B**) Cell counting was performed using a Scepter cell counter (PHCC00000, Millipore) (*n* = 3); * *p* < 0.05: statistically significant difference compared with the control. (**C**,**D**) MTT and LDH assays were performed to assess the metabolic activity of hBM-MSCs after three days of PEMF treatment at various frequencies (*n* = 3); * *p* < 0.05: statistically significant difference compared with the control.

**Figure 2 ijms-23-01177-f002:**
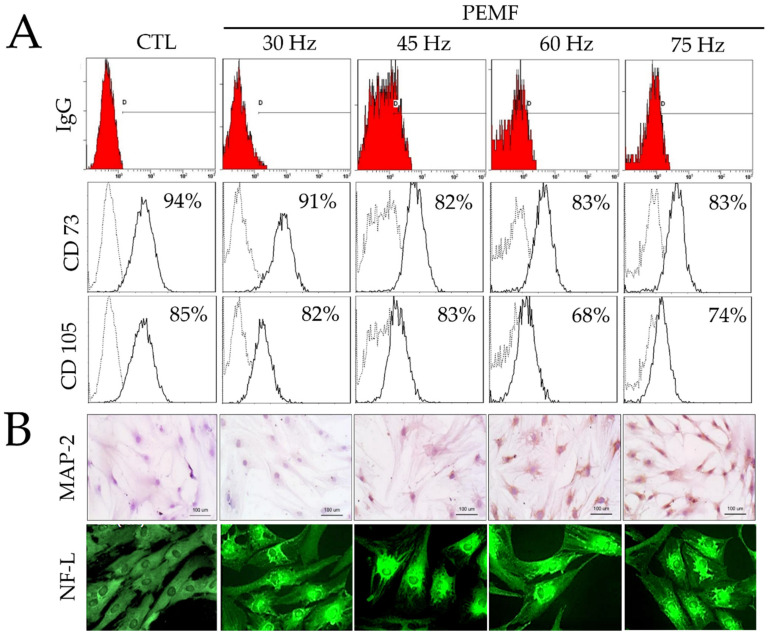
(**A**) Results of fluorescence-activated cell sorting (FACS) analysis on hBM-MSCs surface markers (CD73 and CD105) and IgG control after PEMF for three days. (**B**) Immunohistochemical analysis of MAP-2 antibodies on hBM-MSCs cultured after PEMF for three days (original magnification: 100×); and immunofluorescence staining of NF-L antibodies on hBM-MSCs after PEMF for three days (original magnification: 400×).

**Figure 3 ijms-23-01177-f003:**
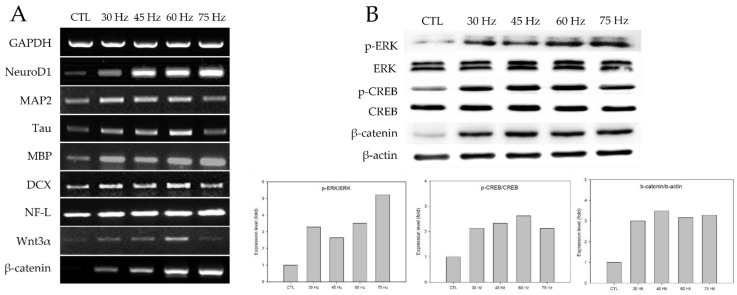
(**A**) Levels of neural gene expression detected by RT-PCR in hBM-MSCs after three days of PEMF treatment. Total cellular mRNA was extracted to conduct RT-PCR. (**B**) Western blot analysis of hBM-MSCs after PEMF for three days. Total cells lysates were immunoblotted with ERK-, p-ERK-, CREB-, and p-CREB-specific antibodies. β-actin served as an internal control.

**Figure 4 ijms-23-01177-f004:**
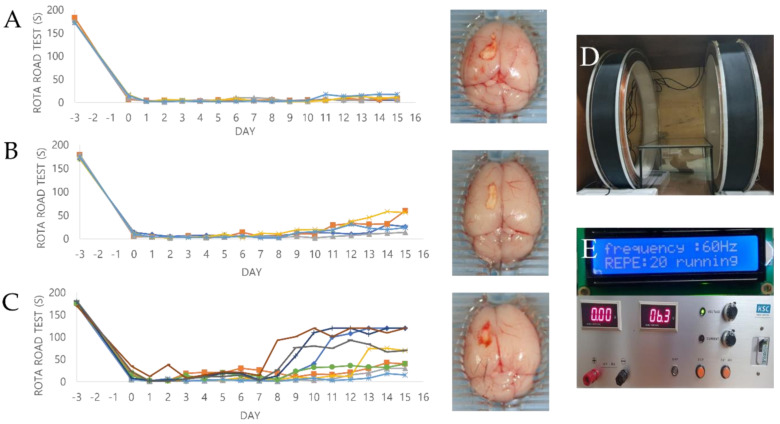
This is the result of the Rota-rod test behavioral experiment for 15 days after the ischemia model. All mice were trained daily for 3 days before the stroke modeling for the Rota-rod (3 cm in diameter, 15 rpm) test. To evaluate animal behavior, the Rota-rod test was performed every day for 15 days after transplantation. −3 days = internal baseline before the surgery, 0 days = right after the surgery. (**A**) Control group (*n* = 5), (**B**) Cell group (*n* = 5), (**C**) Cell/PEMF group (*n* = 9). (**D**) Two solenoid coils. (**E**) Signal generator.

**Figure 5 ijms-23-01177-f005:**
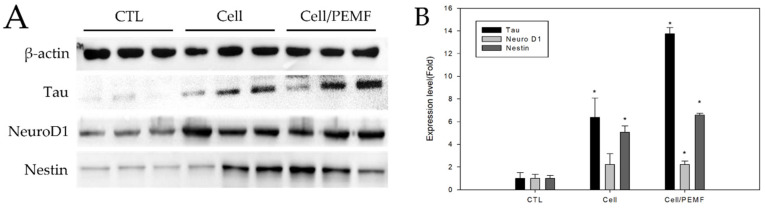
Western blot analysis after transplantation of hBM-MSCs in the cerebral ischemia mouse model at 15 days. (**A**) The cell and cell/PEMF group exhibited higher levels of neural protein expression compared to the control group. (**B**) Quantitative analysis of expression level in comparison to β-actin, an internal control. The Western blot band intensities were measured with the ImageJ software. Each bar represents the mean ± SD of independent experiments performed in triplicate (*n* = 3). * *p* < 0.05: significant difference compared with the control group (*n* = 3).

**Figure 6 ijms-23-01177-f006:**
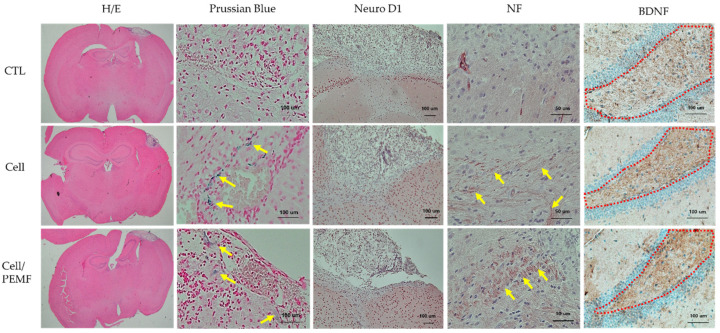
Cerebral infarction area and immunohistochemical staining 15 days after surgery. Expression of neural-related proteins in the control group, cell group, and cell/PEMF group. Yellow arrow: Prussian Blue and NF expression. Hematoxylin and eosin (H&E) staining (original magnification: 1.15×), Prussian Blue (Original magnification: 200×), Neuro D1 (original magnification: 100×), NF (original magnification: 400×), and BDNF (original magnification: 200×).

**Figure 7 ijms-23-01177-f007:**
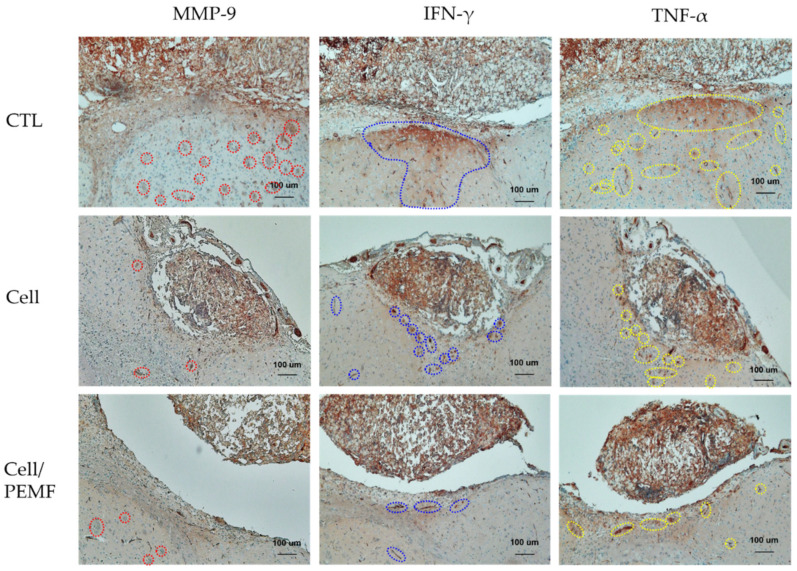
Immunohistochemical staining 15 days after surgery. Expression of MMP-9, IFN-γ, and TNF-α in the control, cell, and cell/PEMF groups. Red dotted line: MMP-9 expression; blue dotted line: IFN-γ expression; yellow dotted line: TNF-α expression (original magnification: 100×).

**Table 1 ijms-23-01177-t001:** Results of the Rotarod Test Conducted for 15 days Using the Stroke Model.

Group	Days						
	9	10	11	12	13	14	15
CTL	3.0 ± 0.94	3.4 ± 1.27	7.6 ± 5.19	8.2 ± 3.49	8.6 ± 4.74	9.6 ± 4.91	9.4 ± 4.70
Cell	11.6 ± 4.79	12 ± 6.30	16.8 ± 8.31	23.6 ± 13.19	24.3 ± 13.57	30.3 ± 16.45	36.0 ± 19.46
Cell/PEMF	31.5 ± 34.51	42.6 ± 44.56	47.0 ± 46.80	53.5 ± 50.16	60.3 ± 45.65	63.3 ± 41.25	64.0 ± 42.93
CTL vs. Cell CTL vs. Cell/PEMF Cell vs. Cell/PEMF	*p* < 0.005 *p* < 0.05	*p* < 0.0005 *p* < 0.05 *p* < 0.05	*p* < 0.005 *p* < 0.05	*p* < 0.005 *p* < 0.05	*p* < 0.005 *p* < 0.005 *p* < 0.005	*p* < 0.005 *p* < 0.005 *p* < 0.05	*p* < 0.005 *p* < 0.005

**Table 2 ijms-23-01177-t002:** Primer Sequences Used for the RT-PCR Experiments.

Gene	Primer Sequences	Length of Amplicon (bp)
GAPDH	5′-ACCACAGTCCATGCCATCAC-3′ 5′-TCCACCACCCTGTTGCTGTA-3′	452
DCX	5′-GGAAGGGGAAAGCTATGTCTG-3′ 5′-TTGCTGCTAGCCAAGGACTG-3′	138
β-catenin	5′-GACACCTCCCAAGTCCTTTAT-3′ 5′-GTACAACGGGCTGTTTCTACG-3′	470
MAP2	5′-CTCAACAGTTCTATCTCTTCTTCA-3′ 5′-CTTCTTGTTTAAATCCTAACCT-3′	401
MBP	5′-GAGGAAGTGAATGAGCCGGTTA-3′ 5′-TTAGCTGAATTCGCGTGTGG-3′	379
NeuroD1	5′-TGAGACGCATGAAGGCTAAC-3′ 5′-GAAATGGTGAAACTGGCGTG-3′	793
NF-L	5′-CAAGAACATGCAGAACGCTG-3′ 5′-GCCTTCCAAGAGTTTCCTGT-3′	376
Tau	5′-AAAGGTGGCAGTGGTTCG-3′ 5′-GGCTGGTGCTTCAGGTTC-3′	138
Wnt3α	5′-TGTTGGGCCACAGTATTCCT-3′ 5′-ATGAGCGTGTCACTGCAAAG-3′	302

## Data Availability

Not applicable.

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
