# Peer review of "Induced Neurodifferentiation of hBM-MSCs through Activation of the ERK/CREB Pathway via Pulsed Electromagnetic Fields and Physical Stimulation Promotes Neurogenesis in Cerebral Ischemic Models"

_ijms, 2022, doi:10.3390/ijms23031177_

Round 1
Reviewer 1 Report
Reviewer Report for Ijms-1525677
The authors present a novel body of work looking at the synergistic effects of BM stem cell therapy combined with pulse electromagnetic stimulation as a therapy for stroke.
- Materials and Methods:
- Line 71: What passage number were the MSCs purchased at?
- Line 77: insert reference for previous publications mentioned
- Line 95: What was the rationale behind the field intensity and length of time chosen?
- Line 112-119:
- MSC Characterization: how did the authors characterize the identity of MSCs? Did the authors label for negative markers? E.g. CD45 or CD11b? Include the data
- Specify the company source for the antibodies used
- What controls were run for the FACS analysis? Were FMO controls included? Please provide control data
- How were the cells cultured prior to transplantation? at what passage number were they administered?
- How were the animals exposed to PEMF?
- Line 153-154: How was the final intensity and length of treatment time chosen?
- Line 173-174: Which area of the brain was the biopsy punch sampled from?
- Line 216-219: what post-hoc tests were performed?
- Line 216-219: what software was used for statistical analyses?
- Results and Figures
- Line 225: ‘PRME’ – should be PEMF
- Figure 1: Can the authors provide clearer images for Fig 1A? the differences between the different intensity images is not very discernable in the current images
- Figure 1: Line 237 – ‘LHD’ should be LDH
- Line 226-227: was there any statistical significance in the reduction of cell numbers? If not, alter wording to indicate trend
- Lines 246-247 and Figure 2A: Indicate the significance for MSC marker reduction across groups.
- Lines 265-280 and Figure 3: Can the authors present quantification of the western data in particular (pERK/ERK for e.g.) – it was mentioned in the methods that this was carried out.
- Figure 4: Do the individual lines represent individual animals? If so, panel B appears to have more than 5 lines. Also, please clarify this in the figure caption. The caption also needs more detail about what the ‘days’ represents. Were there any mortalities in the groups?
- Table 2: Insert a description of what the numbers represent (seconds I assume?), what the ‘days’ represent (days after stroke induction?). Why is data only from Day 9 onwards shown?
- Figure 5: Were the nestin changes not significant when compared to control in both the cell and cell/PEMF groups? Same question for the Tau expression in the cell group.
- Figure 6: The ‘cell’ H&E image appears to be flipped and the tissue damaged – the infarct as evident in the control image (on the RHS) is evident on the LHS in the ‘cell’ image. Please corret
- Figure 6: Prussian blue staining – do the authors have higher mag images for the Prussian blue? Can the authors comment on the existence of these particles at 15 days after injury? Are the nanoparticles ‘transferred’ during differentiation?
- Figure 7: I am not sure of the utility of analyzing cytokine expression data 15 days after injury. In any case, the differences across groups are not very evident in the images presented. Do the authors have some form of quantification for the cytokines? A western blot analysis or a multiplex analysis for example. Also, can the authors specify the areas of the brain presented on the image itself?
- Discussion
- Line 347: ‘infarct cerebral’ – words need to be the other way around
- Line 347-348: Were the MSCs differentiated prior to administration? If not, please formulate the sentence differently.
- Line 409 – data in Table 2 is only from Day 9 on, the significant improvement at Day 7 hasn’t been shown.
- Line 412 – as mentioned in 2g, please discuss mortalities across groups.
Author Response
Dear editor and reviewers
First of all, thank you for giving me the opportunity to revise for this manuscript.
We thank the reviewers for their good comments and advice. We have faithfully corrected as the reviewer’s comments and the statistical processing data is attached as supplementary.
In response to other questions, we have faithfully revised, modified the figure and manuscript.
The results of the first revision are written in blue words. Thank you again for your advice, reviewer.
If our revision is not enough, please comment again. I will try to revise it sincerely.
Best regards
- Materials and Methods:
- Line 71: What passage number were the MSCs purchased at?
Ans) We appreciate your careful reading. We purchased the bone marrow-derived mesenchymal stem cells (PSC-500-012) from ATCC and they stocked the cells in the second passage. We provide the data sheet as evidence.
- Line 77: insert reference for previous publications mentioned
Ans) We appreciate your careful reading. We attached the references and a reference number to the manuscript as you mentioned.
“19. Qian, D. X., Zhang, H. T., Ma, X., et al. Comparison of the efficiencies of three neural induction protocols in human adipose stromal cells. Neurochemical Research. 2010;35(4), 572–579.”
- Line 95: What was the rationale behind the field intensity and length of time chosen?
Ans) We appreciate your careful reading. We added the following explanation.
We performed screening studies with various frequencies and intensities before conducting electromagnetic field studies. In particular, a screening study was tried with an emphasis on intensity and time to maintain survival. Cell research (Fig.B) on 3 hours with 10 mT has been published, but the results according to other concentrations have not been published. Cell survival also differed according to the concentration of Forskolin, and we reduced the exposure time by analyzing and predicting these results.
- Line 112-119: MSC Characterization: how did the authors characterize the identity of MSCs? Did the authors label for negative markers? E.g. CD45 or CD11b? Include the data
Ans) Thank you for your keen question. We purchased and used BM-MSCs from ATCC, and they analyzed CD14, CD19, CD34, and CD45 as a negative control in the characterized data of BM-MSC. Therefore, CD45 was not analyzed in this test. However, we agree with your comment and will proceed with inclusion in future experiments. We provide the data sheet as evidence.
- Specify the company source for the antibodies used
Ans) Thank you for your keen comment. We revised the manuscript accordingly.
“CD 73 monoclonal antibodies (Invitrogen, Product # 12-0739-42, USA) and CD 105 (Endoglin) monoclonal antibodies (Invitrogen, Product # 12-1057-42, USA) were used to detect human antigens.”
- What controls were run for the FACS analysis? Were FMO controls included? Please provide control data
Ans) Thank you for your keen question. We used FMO control and inserted the used IgG data into Figure 2. Also, we found an error in the control data and revised the control data correctly in Figure 2.
- How were the cells cultured prior to transplantation? at what passage number were they administered?
Ans) We appreciate your careful reading. We revised the manuscript accordingly. We used BM-MSCs of passage 5 for transplantation and implanted these cells into mice without inducing neuronal differentiation.
- How were the animals exposed to PEMF?
Ans) We appreciate your keen question. We produced a PEMF system for animal experiments. In addition, an animal experiment was conducted by manufacturing a control device that can control the intensity and frequency of the electromagnetic field. We performed PEMF We attached a figure of the electromagnetic field system used in the animal experiment as follows.
< Image of the PEMF stimulation model: (A) Two solenoid coil, (B) Generator >
- Line 153-154: How was the final intensity and length of treatment time chosen?
Ans) We appreciate your keen question. We found a neural-induced condition of BM-MSC in vitro experiment (10 mT, 60 Hz and 75 Hz, 30 min). Among them, animal testing was conducted under low-frequency conditions for animal safety. We exposed PEMF to animals for 30 minutes at an intensity of 10 mT and a frequency of 60 Hz.
- Line 173-174: Which area of the brain was the biopsy punch sampled from?
Ans) Thank you for your great question. As described in material method 2.8 in the manuscript. We punched the right front core, which is the cerebral infarction part we made (anterior 1.5 mm to the bregma and 2.5 mm lateral from the midline) with a diameter of 3 mm and a depth of 3 mm.
- Line 216-219: what post-hoc tests were performed?
Ans) We appreciate your keen question. We confirmed statistical significance using Tukey’s post hoc of method as a post-test and revised it in the manuscript.
“The results of the animal tests were analyzed using one-way analysis of variance (ANOVA) followed by Tukey’s post hoc test. A P-value < 0.05 was considered statistically significant.”
- Line 216-219: what software was used for statistical analyses?
Ans) Thank you for your great question. We used SPSS 10.0 software as a statistical program and revised it in the manuscript.
“All data were analyzed using the SPSS 10.0 software (SPSS, Chicago, IL) and are reported as mean ± SD. The differences between the groups were assessed using Student’s t-test. The results of the animal tests were analyzed using one-way analysis of variance (ANOVA) followed by Tukey’s post hoc test. A P-value < 0.05 was considered statistically significant.”
- Results and Figures
- Line 225: ‘PRME’ – should be PEMF
Ans) We appreciate your keen point out. We made a mistake and revised the manuscript accordingly.
Figure 1: Can the authors provide clearer images for Fig 1A? the differences between the different intensity images is not very discernable in the current images
Ans) We appreciate your careful reading. We changed the picture of Figure 1A to a high magnification cell picture and revised the manuscript accordingly.
- Figure 1: Line 237 – ‘LHD’ should be LDH
Ans) We appreciate your keen point out. We made a mistake and revised the manuscript accordingly.
- Line 226-227: was there any statistical significance in the reduction of cell numbers? If not, alter wording to indicate trend
Ans) We appreciate your keen comment. We revised the 3.1 of the result in the manuscript as follows.
“The total cell numbers in the treated PEMF groups tended to be lower than that of the control group.”
- Lines 246-247 and Figure 2A: Indicate the significance for MSC marker reduction across groups.
Ans) We appreciate your careful reading. We revised the 3.2 of the result in the manuscript follows.
“Interestingly, CD73 expression tended to decrease at 45, 60, and 75 Hz, whereas CD105 decreased at 60 and 75 Hz compared to the control group.”
- Lines 265-280 and Figure 3: Can the authors present quantification of the western data in particular (pERK/ERK for e.g.) – it was mentioned in the methods that this was carried out.
Ans) We appreciate your keen point out. We added a graph in Figure 3 that quantified the results of the Western blot using Image J, and modified Figure 3 in the manuscript.
- Figure 4: Do the individual lines represent individual animals? If so, panel B appears to have more than 5 lines. Also, please clarify this in the figure caption. The caption also needs more detail about what the ‘days’ represents. Were there any mortalities in the groups?
Ans) We appreciate your keen point out. We made a mistake about panel B and revised the caption of figure 4 in the manuscript accordingly. A total of 10 mice were used for each experimental group, and the animals that died during the experimental period after surgically inducing brain ischemia were excluded from this experiment. Therefore, a total of 5, 5, and 9 mice were ultimately evaluated for the control, cell, and cell/PEMF groups, respectively. The results reported herein correspond to those observed at 9 days, which is when significant differences were observed.
“Results of the rotarod behavioral test 15 days after the surgical induction of ischemic lesions. All mice were trained daily for three days before the surgery in preparation for the rotarod assay (3 cm in diameter, 15 rpm). To evaluate animal behavior, the rotarod test was performed every day for 15 days after transplantation. -3 days = internal baseline before the surgery, 0 days = immediately after the surgery. A: Control group (n=5), B: Cell group (n=5), C: Cell/PEMF group (n=9).”
- Table 2: Insert a description of what the numbers represent (seconds I assume?), what the ‘days’ represent (days after stroke induction?). Why is data only from Day 9 onwards shown?
Ans) We appreciate your keen point out. We revised table 2 and its description in the manuscript accordingly. We measured and evaluated the time(second) spent on Rota-rod before and after surgery. We showed the results after 9 days when a significant difference was observed.
- Figure 5: Were the nestin changes not significant when compared to control in both the cell and cell/PEMF groups? Same question for the Tau expression in the cell group.
Ans) We appreciate your keen point out. We quantitatively analyzed the results of western blot through t-test, and as a result, it was observed that Nestin and Tau had a significant difference compared to the control group. Also, Nestin has observed a significant difference compared to the control group. We revised Figure 5 and its description in the manuscript accordingly.
9.Figure 6: The ‘cell’ H&E image appears to be flipped and the tissue damaged – the infarct as evident in the control image (on the RHS) is evident on the LHS in the ‘cell’ image. Please correct
Ans) We appreciate your keen point out. We arranged the H/E image correctly, and the part where the staining was strongly expressed in the immunostaining result photo was marked with arrows.
10.Figure 6: Prussian blue staining – do the authors have higher mag images for the Prussian blue? Can the authors comment on the existence of these particles at 15 days after injury? Are the nanoparticles ‘transferred’ during differentiation?
Ans) We appreciate your great question. We clearly marked Prussian blue in Figure 5 and revised the description of Figure 5. We agree with your opinion. After 15 days, the injected BM-MSCs will disappear or die by immune cells. However, we judged Prussian blue staining to confirm that the injected cells had moved to the injured area.
11.Figure 7: I am not sure of the utility of analyzing cytokine expression data 15 days after injury. In any case, the differences across groups are not very evident in the images presented. Do the authors have some form of quantification for the cytokines? A western blot analysis or a multiplex analysis for example. Also, can the authors specify the areas of the brain presented on the image itself?
Ans) Thank you for your keen point out. We agree with your opinion. There is no way to quantify immunostaining. Also, since the inflammatory reaction occurs early, it is right to observe the inflammatory reaction within 5 days. However, we performed immunostaining two weeks later to confirm that the inflammation would have a bad effect if it remained for a long time. In future experiments, we will analyze using the ELISA method for protein mass for quantitative analysis. We marked the part where cytokine was expressed as a dotted line in figure 7.
- Discussion
- Line 347: ‘infarct cerebral’ – words need to be the other way around
Ans) We appreciate your keen point out. We made a mistake and revised the manuscript accordingly.
“In this study, we induced neural differentiation by PEMF treatment of hBM-MSC for cerebral infarction healing and demonstrated the effect of electromagnetic fields on post-transplantation healing effects of BM-MSC in mouse ischemic models.”
- Line 347-348: Were the MSCs differentiated prior to administration? If not, please formulate the sentence differently.
Ans) We appreciate your careful reading. In this study, we induced neurodifferentiation by PEMF treatment of hBM-MSC for cerebral infarction healing and demonstrated the effect of electromagnetic fields on post-transplantation healing effects of BM-MSC in mouse ischemic models.
“In this study, neural differentiation of hBM-MSCs was induced via PEMF treatment to assess whether this could serve as a therapy to aid in post-stroke recovery, as well as to demonstrate the effect of electromagnetic fields on the post-transplantation healing effects of BM-MSCs in mouse ischemic models.”
- Line 409 – data in Table 2 is only from Day 9 on, the significant improvement at Day 7 hasn’t been shown.
Ans) We appreciate your keen point out. We made a mistake and revised the manuscript accordingly.
“Additionally, the animals in the cell/PEMF group showed significant improvements nine days after transplantation compared to the control and cell groups, as shown in Table 2.”
- Line 412 – as mentioned in 2g, please discuss mortalities across groups.
Ans) We appreciate your careful reading. In this study, a total of 30 animals were used, and the experiment was divided into control, cell, and cell/PEMF groups. In all experimental groups, one of the mice died immediately after surgery, five in the control group and three in the cell group, during the experiment.

Reviewer 2 Report
Although I am intimately familiar with the field of this manuscript, I found some of the authors’ explanations difficult to follow; I suspect a reader less familiar with the topic might have even greater difficulties. One major concern is that the methods of this work is unclear.
The author should clearly describe the sample-prepared brain region in Western blotting and immunohistochemistry in vivo experiments. In addition, there are some mistakes in the explanation in the results and characters and marks are small and hard to see in the several photograph of figure.
Author Response
Dear editor and reviewers
First of all, thank you for giving me the opportunity to revise for this manuscript.
We thank the reviewers for their good comments and advice. We have faithfully corrected as the reviewer’s comments and the statistical processing data is attached as supplementary.
In response to other questions, we have faithfully revised, modified the figure and manuscript.
The results of the first revision are written in blue words. Thank you again for your advice, reviewer.
If our revision is not enough, please comment again. I will try to revise it sincerely.
Best regards
- Comments and Suggestions for Authors
1.Although I am intimately familiar with the field of this manuscript, I found some of the authors’ explanations difficult to follow; I suspect a reader less familiar with the topic might have even greater difficulties. One major concern is that the methods of this work is unclear.
Ans) We appreciate your careful reading. We revised the methods and figure description in the manuscript.
2.The author should clearly describe the sample-prepared brain region in Western blotting and immunohistochemistry in vivo experiments.
Ans) We appreciate your careful reading. We revised methods in 2.8 and 2.9 in the manuscript.
“A 3 mm diameter and 3 mm depth orifice was punched into the right front core, which is where the cerebral infarction region was made (anterior 1.5 mm to the bregma and 2.5 mm lateral from the midline).”
“For immunohistochemical (IHC) staining, brain sections were incubated with anti-neuroD1, anti-NF, anti-BDNF, anti-MMP-9, anti-IFN-γ, and anti-TNF-α antibodies (1:100; Abcam, Cambridge, UK). The sections were quenched with hydrogen peroxide, unmasked with hyaluronidase, and blocked in 10% normal donkey serum, then incubated with primary antibody for 1 h, washed, incubated with biotinylated secondary antibody, visualized with the Vectastain ABC staining kit (Vector Laboratories, Burlingame, CA, USA), and developed in 3,3-diaminobenzidine.”
- In addition, there are some mistakes in the explanation in the results and characters and marks are small and hard to see in the several photograph of figure.
Ans) We appreciate your careful reading. We revised the result and figure description in detail in the manuscript. Also, we revised marks of figures in the manuscript.

Round 2
Reviewer 2 Report
The manuscript has been improved and is in a nice condition now.